# The Role of N^6^-Methyladenosine in the Promotion of Hepatoblastoma: A Critical Review

**DOI:** 10.3390/cells11091516

**Published:** 2022-04-30

**Authors:** Finn Morgan Auld, Consolato M. Sergi, Roger Leng, Fan Shen

**Affiliations:** 1Department of Laboratory Medicine and Pathology, University of Ottawa, Ottawa, ON K1N 6N5, Canada; fauld@toh.ca; 2Division of Anatomical Pathology, Children’s Hospital of Eastern Ontario (CHEO), Ottawa, ON K1H 8L1, Canada; 3Department of Laboratory Medicine and Pathology, University of Alberta, Edmonton, AB T6G 2B7, Canada; rleng@ualberta.ca

**Keywords:** hepatoblastoma, N^6^-Methyladenosine, m^6^A, beta-catenin, methyltransferase, hepatoblastoma genetics

## Abstract

**Simple Summary:**

Hepatoblastoma is the most common malignant pediatric tumor of the liver. Unlike hepatocellular carcinoma (HCC) which has been associated with hepatitis B virus infection or cirrhosis, the etiology of hepatoblastoma remains vague. Genetic syndromes, including familial adenomatous polyposis (FAP), Beckwith-Wiedemann syndrome (BWS), and trisomy 18 syndrome, have been associated with hepatoblastoma. BWS is an overgrowth syndrome which exhibits an alteration of genomic imprinting on chromosome 11p15.5. N^6^-Methyladenosine (M^6^A) is an RNA modification with rampant involvement in the metabolism of cells and malignant diseases. It has been observed to impact the development of various cancers via its governance of gene expression. Here, we explore the role of m^6^A and its genetic associates in promoting HB, and the impact this may have on our future management of the disease.

**Abstract:**

Hepatoblastoma (HB) is a rare primary malignancy of the developing fetal liver. Its course is profoundly influenced by genetics, in the context of sporadic mutation or genetic syndromes. Conventionally, subtypes of HB are histologically determined based on the tissue type that is recapitulated by the tumor and the direction of its differentiation. This classification is being reevaluated based on advances on molecular pathology. The therapeutic approach comprises surgical intervention, chemotherapy (in a neoadjuvant or post-operative capacity), and in some cases, liver transplantation. Although diagnostic modalities and treatment options are evolving, some patients experience complications, including relapse, metastatic spread, and suboptimal response to chemotherapy. As yet, there is no consistent framework with which such outcomes can be predicted. N^6^-methyladenosine (m^6^A) is an RNA modification with rampant involvement in the normal processing of cell metabolism and neoplasia. It has been observed to impact the development of a variety of cancers via its governance of gene expression. M^6^A-associated genes appear prominently in HB. Literature data seem to underscore the role of m^6^A in promotion and clinical course of HB. Illuminating the pathogenetic mechanisms that drive HB are promising additions to the understanding of the clinically aggressive tumor behavior, given its potential to predict disease course and response to therapy. Implicated genes may also act as targets to facilitate the evolving personalized cancer therapy. Here, we explore the role of m^6^A and its genetic associates in the promotion of HB, and the impact this may have on the management of this neoplastic disease.

## 1. Introduction

Hepatoblastoma (HB) is a rare and heterogeneous malignancy deriving from embryonic liver parenchyma. Despite affecting fewer than two children in a million, it represents over a third of primary tumors of the pediatric liver [1,2,3,4]. Its genesis and progression are highly genetically mediated, indicated by sporadic forms adjacent to a strong syndromic association, but also intimately associated with the frequent occurrence of extramedullary hematopoiesis, as identified in early studies on the developing intrahepatic biliary system of the fetal liver [5].

Subtypes are described histologically, based on the ratio of epithelial to mesenchymal tissues present and their stage of aberrant differentiation [6]. Fetal-type tumors are further differentiated than their more immature embryonal counterparts. Due to the architectural and cytological complexity of tumors, and compounded by their rarity, consensus between experts is not always established. This is particularly observed amongst subtypes such as small cell undifferentiated, which unfortunately traverses a less favorable clinical course [2,4,6]. The 5-year survival rate of patients with HB approaches 80% due to advances in early detection and multidisciplinary therapy, however, it remains challenging to predict the clinical course. Relapse of disease, metastatic spread, and variable response to chemotherapy are grave concerns that affect many patients [4,7].

N^6^-Methyladenosine (m^6^A) is an extraordinarily prevalent modification of eukaryote RNA [8,9]. Its presence governs multiple facets of normal cellular processing, differentiation, and cell cycle regulation. It is widely implicated in malignant diseases [10].

Epigenetics has been recognized as a novel concept pinpointing the epigenetic trait as a firmly heritable phenotype, which arises from modifications in a chromosome devoid of alterations in the DNA sequence. There are more than one hundred chemical modifications (of which m^6^A is one) that have been identified on RNAs. Such changes harbor significant biological functions in living organisms using their involvement in intervening in epigenetic regulation. Such a role has been detected mainly in the origin and progression of hematologic malignancies [11]. At present, epitranscriptomics, represented by m^6^A changes, has grown into a major research topic. This development is in part linked to second-generation sequencing tackling epigenetic modification at the transcriptome level. At its core, m^6^A methylation changes can arbitrate the post-transcriptional control of gene expression devoid of altering base sequences. In adding together to this concept, RNA m^6^A change is (1) reversible and (2) dynamically modulated by modifying molecules or modifiers (e.g., writers, readers, and erasers). They have presently been demonstrated to play a crucial role in regulating not only mRNA decay and stability, but also splicing, translation efficiency, and localization. Further, m^6^A sites have also been recognized in long non-coding RNAs as well as non-coding RNAs, such as microRNAs. The development of high throughput m^6^A sequencing technology has enormously accelerated in recent years. There is accumulated evidence that m^6^A and its related factors are involved in HB [12,13,14]. It seems that the regulation consists of the self-renewal, proliferation, and differentiation of the neoplastic cell.

Some antibody-dependent (such as methylated RNA immunoprecipitation sequencing (MeRIP-seq) and m^6^A individual-nucleotide-resolution cross-linking and immunoprecipitation (miCLIP)) and antibody-independent (such as MAZTER-seq deploying a sequence-specific, methylation-sensitive, single-stranded ribonuclease MazF, m^6^A-sensitive RNA-endoribonuclease–facilitated sequencing, and deamination adjacent to RNA modification targets sequencing or DART-seq) are terrific tools for detecting and measuring RNA modifications [15]. Next-generation sequencing (NGS) methods have been developed for m^6^A sequencing, making high-resolution detection of m^6^A epitranscriptomes in diverse cell contexts a reality [16]. Indeed, the advent of such NGS-based methods advanced our understanding of this epigenetic marker.

Here, we review the advancement of research on the biological characteristics of m^6^A methylation in HB. We hope to provide a basis for developing molecularly targeted therapies established on the aberrant m^6^A modifications in potentially related hematologic neoplasms.

## 2. The Form and Function of m^6^A

Modifications frequently occur within the mRNA of eukaryotic organisms and have been implicated in a plethora of normal cellular processes, as well as in the governance of cancers [10,17]. The most numerous of these modifications is m^6^A, a post-transcriptional methylation occurring at the N^6^-position of the adenosine base. In particular, m^6^A is highly conserved across normal body tissues and in malignant cell populations, implying its durability and stability, as well as its wide-reaching function. Simply put, it affects every aspect of RNA metabolism [10].

The process of m^6^A methylation is dynamic and reversible, akin to other forms of epigenetic regulation, supporting the proposal that it falls under this umbrella. The formation of m^6^A occurs through a methyltransferase complex comprising a methyltransferase-like (*METTL*) 3 and *METTL14* heterodimer in a catalytic core and a multi-protein regulatory subunit, prominently featuring Wilms’ tumor 1-associated protein (*WTAP*) [12,18]. The effects of m^6^A are breathtaking in their diversity; fittingly, they are mediated by an expanding list of biological components, broadly grouped into readers, writers, and erasers according to the way in which they interact with m^6^A (Figure 1).

The behavior of certain cancers, including their genesis and progression, has long been linked to the expression of tumor-related genes. In this context, the modulation of heritable gene expression without DNA sequence changes is often stimulated by alteration of m^6^A levels [8]. The sequence in which m^6^A methylation takes place is catalyzed by a host of genes that make up the writer complex, an elaborate conversation in which substrates are installed into the m^6^A methylation prior to the determination of downstream RNA fate by other reader proteins [9]. Regulators of m^6^A RNA methylation are involved in various human diseases through the effects they exert and thus provide potential targets for cancer therapy [8].

Characterization of the genetic and epigenetic promotion of rare malignancies such as HB is of great clinical importance. Histologic subtyping suggests that some patterns experience worse outcomes than others, from aggressive growth and early spread to resistance to current chemotherapeutic mechanisms. Poorly differentiated subtypes, such as the small cell undifferentiated histologic subtype, represent a diagnostic and management challenge due to their rarity and recalcitrant behavior. The acquisition of mesenchymal features, even in small quantities, correlates with chemoresistance [1]. Troublesome subtypes consistently show the arrest of normal cellular development at an earlier embryologic stage [19]. Processes of genetic modification that impact the differentiation of embryologic tissues, such as those governed by m^6^A, have been of particular interest. In addition to identifying therapeutic targets, understanding the mechanism by which the disease propagates may identify those who are predisposed outside of the established genetic syndromes, stratify risk, and offer clarity around patient outcomes [7,8,10].

The m^6^A methylation processing and its biological functions is depicted in Figure 1, Figure 2, Figure 3 and Figure 4. Figure 1 displays m^6^A writers (methyltransferase) methylating RNA in the adenine nucleobase of the amino group at N^6^ position. The consensus motif of *METTL3* is RRA*CH (R = A/G; A* = methylated A; H = A/C/U). Once m^6^A is deposited on RNA, m^6^A reader proteins are recruited and determine RNA fates, such as splicing, stability, and translation efficiency, ultimately affecting gene expression. Also, m^6^A is removed from RNA through demethylation by eraser proteins.

Zinc finger CCHC domain-containing protein 4 (*ZCCHC4)* possesses several zinc finger motifs, including Gly-Arg-Phe (GRF), Cys2-His2 (C2H2), and Cys-Cys-His-Cys (CCHC) domains. Methyltransferase-like 5 (METTL5), acts as a conserved methyltransferase, specifically catalyzing m^6^A at the *18S* A1832 motif. METTL5 allows S-adenosyl-L-methionine binding activity in addition to rRNA (adenine-N6-)-methyltransferase activity. METTL5 is involved in the positive regulation of translation and rRNA methylation [20]. Fat mass and obesity-associated protein (FTO) and alkb homolog 5 (ALKBH5) contain KGFe(II)-dependent dioxygenase domains conserved in dioxygenase family enzymes.

FTO’s role as an m^6^A eraser is controversial since several studies showed that FTO is solely responsible for the demethylation of m^6^A on snoRNA 5′ cap [21,22,23]. Essentially, FTO preferentially demethylates m^6^Am (N6,2′-O-dimethyladenosine) detected in 5′-caps of mRNAs, rather than m^6^A [23]. Several m^6^A detection technologies unintentionally map m^6^Am changes as m^6^A sites. Overall, the thought of reversible m^6^A remains controversial. The biological role and evidence of its removal are still poorly understood. It, obviously, adds a layer of complexity when trying to associate m^6^A with neoplastic progression. Dominissini et al. introduced the human and mouse m^6^A modification landscape in a transcriptome-wide manner based on antibody-mediated capture and parallel sequencing [24]. They identified over 12,000 m^6^A sites showing a typical consensus in the transcripts of more than 7000 human genes. Two distinct landmarks appeared, the one around stop codons and the other within long internal exons. Both are highly conserved between humans and mice [22,23,24,25,26,27]. Additionally, Meyer et al. found that m^6^A is massively enriched in the three prime untranslated regions (3′ UTRs) as well as near the stop codon in mature mRNA [28]. Figure 2 displays that *METTL16* methylates a stem-loop structure in the 3′ untranslated region (UTR) of S-adenosyl methionine (SAM) synthase, methionine adenosyltransferase 2A (*MAT2A*). In SAM-repleted conditions, *MAT2A* is methylated and degraded. Conversely, in SAM-depleted conditions, *METTL16* induces splicing and expression of *MAT2A*. Figure 3 displays the methylation of A4220 in 28S ribosomal RNA (rRNA), *ZCCHC4* promotes ribosome assembly and translation.

Figure 4 exhibits the domain composition of m^6^A enzymes. In addition, m^6^A writers contain methyltransferase (MTase) domains. *METTL3* contains CCCH zinc finger motifs. *METTL16* has two vertebrate-conserved regions (VCR) or domains.

## 3. Key Genetic Players in Hepatoblastoma

Many cytogenetic changes have been cataloged in HB. Most of these changes result from highly penetrant somatic mutations in undifferentiated or minimally differentiated fetal hepatocyte precursors. Such modifications often involve the Wnt signaling pathway, influencing the fate of the transcriptional cofactor beta-catenin [29,30,31]. Beta-catenin is encoded by *CTNNB1*, a proto-oncogene with a pivotal role in cell-to-cell communication and adhesion [12,32].

As a critical component of the Wnt/beta-catenin signaling pathway, *CTNNB1* is mutated in 50–90% of HB [12]. *CTNNB1* is consistently upregulated in HB tumor cells and, when knocked down, reduces their viability and induces apoptosis [12]. Furthermore, there is a statistically significant correlation between the expression of *CTNNB1* and *METTL3*, which speaks to an intimate relationship between the two components. The m^6^A modulation of *METTL3* in turn regulates the processing of *CTNNB1*, sharing its responsibility in driving HB [32]. *TERT* is a regulator of telomerase, an enzyme related to cell immortality, and a significant player in Wnt signal activation [33]. Its expression is enhanced by *MYC,* which is itself an activated target gene of Wnt signaling. In this way, a vicious cycle can occur, by which upregulation of the Wnt signaling pathway further stimulates its drivers [34].

*METTL3* is implicated in malignant disease and normal physiological processes, such as regulating hematopoietic stem cell differentiation [18]. In addition to its function as the catalytic core of the m^6^A writer complex, *METTL3* can act explicitly as an oncogene, promoting HB proliferation while spurring tumor growth by virtue of its role in m^6^A modification [12,18].

Much like other genes involved in the modification of m^6^A, *WTAP* exhibits diverse and complex biological functions. It has been established as tumorigenic in many cancers, including hepatocellular carcinoma (HCC) and osteosarcoma, particularly, through its impact on m^6^A [35,36]. In the governance of normal embryological development and cellular processing, *WTAP* is a key player, engaging in transcriptional and post-transcriptional regulation of cellular genes to guide the organization of biological structures such as the genitourinary system and in various facets of cell-cycle regulation [37,38]. Intrinsic to normal biology and pathology alike, *WTAP* demonstrates a similar duality to m^6^A itself and many of its associated genes.

Regarding its role as a risk factor for HB, Zhuo et al. propose that single nucleotide polymorphisms of *WTAP* potentially predispose to HB development, acting in this context as a genetic modifier [7]. Upregulated expression levels of the *WTAP* gene appeared to increase the risk of developing HB in children. Certain genotypes were observed to modify the risk in specific sub-populations: children under seventeen months of age, female children, and progression in those with stage I or II tumors [7].

The YT521-B homology (YTH) domain family members, including *YTHDC1* and *YTHDF2*, are widespread and highly conserved within eukaryotic cells [35]. They function to specifically bind m^6^A and mediate its interaction with mRNAs; this established function places them under scrutiny for their role in m^6^A dependent cancers [18]. However, whilst each has been linked to pathological processes in diverse body tissues, their contribution to the development of HB has only recently been investigated.

*YTHDC1* is a nuclear m^6^A reader, regulating RNA splicing in a concentration-dependent manner [35]. Its role in mRNA processing has been implicated in numerous cancer-driving processes, such as angiogenesis, growth factor signaling, metastatic spread, apoptosis, and genetic instability [35]. Polymorphisms of m^6^A-associated genes play an essential role in HB, and preliminary investigation suggests that *YTHDC1* is no different [32]. While polymorphisms selected by Chen et al. did not contribute to HB susceptibility, stratification analysis within the same study did detect the potential contribution of *YTHDC1* to HB risk [35,39]. This finding begs further study in a larger cohort to tease out additional polymorphisms of interest.

*YTHDF2* also functions as a crucial reader, usually modifying the degradation of m^6^A-modified mRNAs [39]. The study of its relationship to HB is in its infancy. Still, preliminary work by Cui et al. suggests that its role in developing HB may confer unfavorable clinical outcomes [13].

## 4. The Impact of m^6^A on Tissue Development

The proliferation of HB is governed by the aberrant differentiation of hepatocytes in early life [6,29]. Whether malignant transformation occurs during the earliest phases of differentiation or through gene mutations that impact an already differentiated tissue, abnormal methylation is ubiquitous in implicated genes. The disseminated effects of losing m^6^A modification agents, such as *METTL3*, impact the developmental stages of a variety of tissues [8]. Loss of *METTL3* in animal models, such as in the murine fetal liver and in zebrafish embryogenesis, can activate downstream signaling pathways in hematopoietic stem cells and progenitor cells and influence Notch-dependent signaling [18].

Specifically, m^6^A methylation exhibits a dynamic pattern of downstream effects in the developing liver. To catalogue effects and pursue an underlying mechanism, a transcriptome-wide investigation of porcine liver at different developmental stages has been performed, emphasizing the relationship between m^6^A methylation and gene expression^6^. Genes involved with hepatocyte differentiation and liver development are consistently highly m^6^A methylation dependent. Interruption of the normal function of this process can reasonably be suspected in many cases in which fetal hepatocytes undergo aberrant development [40].

## 5. Promotion of Hepatoblastoma via the m^6^A Pathway

The promotion of cancer via m^6^A modification is achieved through various mechanisms. As we have described, the biological effects of m^6^A are diverse and associated with normal eukaryote physiology as much as they are implicated in disease. When physiological processes such as DNA damage responses and pluripotency deviate from normal, aberrant m^6^A modification is often the culprit [41]. The m^6^A RNA methylation promotes cancer via two overarching roles: a tumor promotor or a suppressor of innate anti-tumor mechanisms. An example of the former is the modification of m^6^A methylation by *METTL3* in its role as a proto-oncogene, enhancing m^6^A modification and degradation of *SOCS2* with the subsequent development of HCC. As a suppressor of protective biological functions, *METTL3* downregulation by m^6^A modulates in the context of endometrial cancer, leading to increased cell proliferation and tumorigenicity [28]. Other familiar characters come to light when investigating the clinical significance of genes associated with m^6^A modification in a pathological context, chiefly: *CTNNB1*, *METTL3*, *WTAP*, *YTHDC1,* and *YTHDF2* [12,13]. Preliminary studies have identified polymorphisms in several of these genes, however, such studies are bound by confounding factors that must be acknowledged: firstly, sample sizes are often small due to the rarity of HB in all populations, a problem that is amplified when subgroups are stratified. Secondly, it can be problematic to extrapolate from genetic studies in isolation. Finally, HB is a profoundly heterogeneous disease and therefore should be considered alongside environmental factors. As a nod to its biological heterogeneity, m^6^A does not function solely to propagate cancer. Instead, it demonstrates a more nuanced dual role, in some instances inhibiting tumor progression. For example, in glioblastoma multiforme (GBM), the downregulation of m^6^A leads to decreased levels of the *ADAM19* gene, enhancing its expression [10]. The enhanced expression of this gene promotes cellular self-renewal and tumorigenesis. A similar phenomenon is observed in *METTL3* within animal embryologic models, in which knockdown of *METTL3* is associated with reduced self-renewal^10^. HB exhibits this complex relationship well, owing to its propagation of a flurry of m^6^A-mediated promotions and cellular circuit breaks. The overexpression of m^6^A has been a ubiquitous observation in HB. Since the Wnt/beta-catenin signaling pathway primarily drives HB, it can be reasonably extrapolated that upregulation in m^6^A causes tumor growth by influencing genes common to this system. Liu et al. showed that genes such as *CTNNB1*, *CCND1,* and *NKD1* showed increased m^6^A methylation compared with non-tumor tissues [12]. There is a consistently observed tendency for m^6^A modification to aggregate in HB, emphasizing its role in the malignant process. *METTL3* is frequently identified as a critical player in HB development machinery [32]. In addition to its impact, *METTL3* influences *CTNNB1* as a downstream target. *CTNNB1* is an infamous proto-oncogene encoding beta-catenin, a protein mutated in most HB cases [4]. In this context, the mutated form of beta-catenin is degradation resistant, accumulating in the nucleus of hepatocyte precursors and binding to the transcription factor TCF4/LEF-1; in doing so, it activates target genes such as *c-MYC* and *Cyclin-D1* [42]. The impaired degradation of beta-catenin and its associated impact on downstream genes is a common theme in proposed mechanisms of HB development. Loss of function somatic mutations in the tumor suppressor genes *AXIN1* and *AXIN2* impair beta-catenin degradation, augmenting the above carcinogenic pathway [43,44].

The workhorse of m^6^A formation is the *METTL3/METTL14* heterodimer and its regulatory subunit, which features WTAP in prominence [21,45]. The genesis and propagation of HB are linked to abnormal m^6^A modification. An elevated level of m^6^A is identified in tumor cells, alongside upregulation of a familiar cohort of genes, including *METTL3*, *WTAP*, etc. *FTO* and *YTHFD2* [8,45,46,47,48]. Interestingly, no upregulation of *METTL14* is identified within tumor cells compared with background liver [8]. Liu et al. investigated the functional role of m^6^A affiliated genes in HB. Knockdown of the above mentioned, overexpressed genes markedly suppressed HB cell proliferation and caused HB cells to undergo apoptosis [12]. This translates clinically to a reduction in tumor size and weight, when *METTL3* is knocked out stably in a murine model^10^. In keeping with the sequence of downstream effects of *METTL3* in vitro, the transcription factor of *CTNNB1* (TCF4/LEF-1) and target gene (*Cyclin-D1*) were downregulated in the knockout murine HB cells [12]. From this observation, it is not too far a stretch to propose the downregulation of *CTNNB1* itself and subsequently of the beta-catenin product in vivo, in a *METTL3* knockout model. *METTL3* has been isolated as the main factor in aberrant m^6^A modification in the context of HB, above that of *WTAP*, its co-catalytic associate. The mechanism by which *WTAP* influences the development and progression of HB itself remains undetermined. Zhuo et al. highlight the ability of certain single nucleotide polymorphisms to enhance neighboring critical genes. For example, *TCP1* is enhanced by a functional genotype of *WTAP* and is known to be implicated in several solid organ malignancies. The door is open for the study of additional single nucleotide polymorphisms of *WTAP*, which may be of functional significance.

## 6. The Role of m^6^A Associated Genes as Diagnostic and Prognostic Biomarkers

There is a significant correlation between an increased protein expression level of m^6^A modifiers and HB. This prompts important clinical questions regarding the utility of associated genes as diagnostic and prognostic biomarkers. Cui et al. illustrate the function of *METTL3* as an oncogene in HB, regulated by the micro-RNA miR-186, to propagate HB via the Wnt/beta-catenin signaling pathway [13]. The axis of interaction between *METTL3* and its micro-RNA modifier may represent a therapeutic target or prognostic biomarker for patients with HB. In addition, polymorphisms of *METTL3* may affect the occurrence of HB; this idea is extrapolated to suggest that certain genotypes can incur a greater risk of HB development [12,32]. Prediction of susceptibility and clinical course is of great importance for a disease in which early detection and treatment correlate strongly with survival. Complementing this, Liu et al. explored the relationship between *METTL3* expression and the clinicopathological characteristics of patients with HB. Increased *METTL3* expression was associated with frequent recurrence and poor survival [32]. Increased expression of *YTHDF2*, the m^6^A reader of a different mechanism, is also significantly associated with poor overall survival rates in HB [13]. These observations add to existing histologic descriptions, forming a helpful constellation of features that predict poor outcomes. While undoubtedly in its early stages, such research offers promise for the diagnosis, prognosis, and risk stratification of susceptible or existing patients^10^. Early exploration of single nucleotide polymorphisms of *WTAP* hint at the potential to discover additional functional genotypes, which may elucidate the mechanisms that drive HB. This understanding will enable a more accurate catalog of mutation sequences in the evolution of HB. It will perhaps inform how aberrant *METTL3* expression is understood under their shared role in m^6^A modification. The identification of HB with high malignant potential is a step towards personalized, precise management, supplementing clinical predictors such as alpha-fetoprotein level (AFP), age at diagnosis and pre-treatment extent at disease (PRETEXT) classification [49]. Description of the pathogenesis of HB and the illumination of key genetic players has not always correlated with tumor behavior and risk stratification, which is of immediate utility to clinicians and patients. Meaningful correlation has been hampered by small patient sample sizes and the sheer range of molecular and genetic targets analyzed. Nagae et al. describe a relationship between age at diagnosis, and the vista of known genetic aberrations [49]. A changing pattern of gene mutation was identified as the age at diagnosis increased. Amongst patients younger than eight years old, mutation of *CTNNB1* formed the overwhelming majority [49]. Patients in which *TERT* is frequently mutated tend to be older, approaching young adolescence [49]. *TERT* mutation has been implicated in particularly aggressive tumors, which is in keeping with the increase in discordant tumor behavior with advancing age at diagnosis [49,50,51]. The changing genetic landscape of tumors with patient age also raises the possibility of discrete tumor biology governing clinical expression [49].

The idea that tumor biology may vary according to clinical behavior is further supported by the description of different epigenetic hallmarks within the HB cohort [52]. Such hallmarks are qualified by the degree of hyper- or hypomethylation of DNA, leading to discriminable effects on the transcriptome [52]. Since the transcriptome influences the molecular behavior of the tumor, it follows that teasing out such epigenetic profiles may be of value to prognostication and risk stratification [52]. An example of this idea in practice is the repression of tumor suppressors in aggressive HB, by defined epigenetic hypermethylation: CpG island hypermethylation. By extrapolating advances in the understanding of HB epigenetics, Carrillo-Reixach et al. proposed the utility of a molecular risk stratification system, which may augment existing clinical models [52].

Moving beyond prognostication to suggest mechanisms of treatment, genes implicated in the immune microenvironment of tumors are of particular interest. The tumor suppressor gene *AXIN1*, alongside other aberrantly methylated genes *LAMB1* and *NOTUM*, were deemed by Zhang et al. to be closely related to immune cells in hepatoblastoma tissues, in which they were upregulated [43]. This finding is of great potential utility as *NOTUM* knockdown subsequently attenuated growth, migration, and invasion of tumor cells in a murine model [43]. Inhibition of the gene *CHKA*, involved in membrane biosynthesis in the context of normal cellular function, has arrested tumor growth in a selection of tumor types, including HB, HCC, breast, lung, and prostate [52]. This finding emphasizes the implicitly shared pathways in tumorigenesis across different body sites and suggests *CHKA* as a target for future HB treatment [52]. As aberrantly methylated and expressed genes are cataloged in droves, their potential therapeutic roles can be investigated by describing the interplay between genes and other cancer-modulating mechanisms such as the host immune response.

Recently, specific inhibitors of both m^6^A eraser and writers have been developed. The design of 20 molecules with low micromolar IC50′s and specificity toward FTO over ALKBH5 confirmed two competitive inhibitors, FTO-02 and FTO-04. Remarkably, FTO-04 prevented neurosphere formation in patient-derived glioblastoma stem cells (GSCs) without preventing the growth of healthy neural stem cell-derived neurospheres. FTO-04 increased m^6^A and m^6^Am levels in GSCs consistent with FTO inhibition as well. This data may quite strongly support FTO-04 as a potential new lead for the therapeutic handling of glioblastoma [53]. Besides, other FTO inhibitors such as FB23/FB23-2, MA/MA2, MO-I-500, and, intriguingly, Entacapone have been delineated to harbor a binding ability toward FTO with the aim to reduce the RNA demethylase activity, which should have anti-cancer effects other than anti-obesity effects [54]. Also, METTL3 inhibitors and FTO inhibitors have been proposed to harbor an immunotherapeutic potential in controlling acute myeloid leukemia [55].

## 7. Detection of RNA Modification by Nanopore Sequencing

There are several methods to detect RNA modifications, including antibody immunoprecipitation (e.g., MeRIP-seq, miCLIP) [56,57] and chemical-based modification, required to convert RNA to complementary DNA (cDNA). It has been noted that cDNA-based methods through reverse transcription or amplification are vulnerable to bias [58]. These concerns can be aggravated by the concurrent use of old-fashioned short-read sequencing technologies, which are known to exhibit GC bias. These technologies rely on available antibodies or known enzymes, such as ligases, which indicate a preference for specific sequences or nucleotides [56]. These techniques are often unable to identify the underlying RNA molecule that is modified, for example, base modifications that are known to have a role in controlling the activity and stability of RNA [58]. Thus, the large-scale application of these technologies is challenged by the requirement for complex protocols.

To address these limitations, a direct RNA sequencing platform provided by Oxford Nanopore Technologies (ONT) is proposed as a substitute technology to determine sites of alteration in the native RNA molecule [58]. Direct RNA nanopore sequencing has been used to analyze m^6^A in Arabidopsis [59], yeast [58,60], RNA virus genomes [61], and human cells [62,63,64,65,66]. In particular, Workman et al. focused on the m^6^A methyltransferase-binding motif, a short recurring pattern in its architecture with presumed biological function [64]. An ionic current change was demonstrated within the motif, attributable to m^6^A. This signal difference was subsequently validated utilizing data from the synthetic RNA. Fascinatingly, m^6^A-modified motifs in isoforms of the same gene were identified by using the ionic current difference. Whilst Workman et al. describe a relatively higher error rate of ONT when compared with cDNA sequencing, this technology directly detects post-transcriptional modifications, providing valuable insight into the mechanics of associated disease [64].

More recently, a computational method known as xPore has been developed by Pratanwanich et al. [62]. This technique enables differential RNA modifications from direct RNA sequencing data to be retained. This technology was tested on direct RNA sequencing data across six genetically distinct human cell lines covering liver cancer cells (HEPG2), colon cancer cells (HCT116), breast cancer cells (MCF7), lung adenocarcinoma cells (A549), and leukemia cells (K562), and HEK293T-KO cells. Between 800 and 2000 differentially modified sites were identified for all five cancer cell lines; most sites conformed to m^6^A DRACH motif. These findings indicated that RNA modifications could be observed across conditions, even when samples have a diverse genetic background. In the same study, the authors identified the dynamics of m^6^A by investigating the different tissues represented by the cell lines. Profoundly, the authors found that many m^6^A sites are preserved across multiple cell lines, with most positions being shared [62]. A significant advantage of xPore is that it is suited for detecting m^6^A with direct RNA-seq data from clinical cancer samples, even in the context of limited RNA (e.g., 2.5 micrograms) [62]. Since xPore does not rely on strict case-control comparisons, it offers essential flexibility in analyzing primary tissues and patient samples. False-positive results are primarily avoided by stringent filtering, which, in conjunction with the large data handling capacity of the method, opens new opportunities for larger-scale analysis of clinical patient data [62]. Furthermore, Jenjaroenpun et al. employed native RNA sequencing on lung cancer cell lines H460 and CEPH1463 [66,67,68]. They detected m^6^A in the RNA using epitranscriptomal landscape inferring from glitches on ONT signals (ELIGOS), in which caught error signatures are used to elucidate the properties of RNA. While it accurately determines RNA modifications’ specific locations, ELIGOS technology relies upon other sequencing methods to provide the proxy of detection. Therefore, traditional sequencing methods are required to understand the nature of the modification ELIGOS reveals [66]. Although still in its infancy, direct RNA sequencing has the propensity to detect RNA base modifications. Nanopore sequencing and its adjuncts such as ELIGOS will serve to update our current knowledge of the epitranscriptome in cancer.

## 8. Conclusions

HB represents the majority share of a rare disease cohort, with an incidence that appears to be increasing. Abnormal activation of a constellation of genes occurs to govern its inception and clinical course, many of which are linked to aberrant methylation of RNA. Although the investigation of HB’s genetic and epigenetic basis is in its relative infancy, there is a healthy precedent linking these genes to the trajectory of other solid organ malignancies. In keeping with its behavior in other biological systems, m^6^A appears to be a centerpiece in the complex genetic promotion of HB. Sequencing technologies and their adjuncts serve to elucidate the location and nature of m^6^A modifications within RNA and reveal the scale of their role across human tissues and diseases. The future will undoubtedly shed further light on the consequences of this modification for the patient journey. In this way, predictive and management decisions about the individual patient will rightly remain at the forefront of consideration.

## Figures and Tables

**Figure 1 cells-11-01516-f001:**
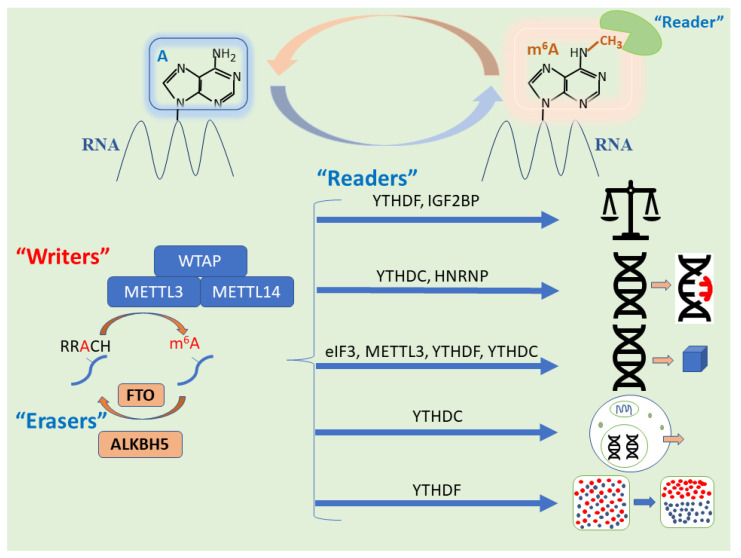
The m^6^A methylation is catalyzed by the writer complex, including METTL3, METTL14, METTL16, WTAP. The m^6^A modification is erased by demethylases, including FTO and ALKBH5, where FTO is preferentially responsible for the demethylation of N^6^,2′-O-dimethyladenosine (m^6^A_m_) at the 7-methylguanosine cap. The m^6^A-modified RNA reader proteins include YTHDF1/2/3, YTHDC1/2, IGF2BP1/2/3, and HNRNPC/A2B1. M^6^A modification modulates miRNA biogenesis, inactivation, m^6^A switch, RNA translocation, pre-mRNA splicing, RNA translation, RNA decay, and RNA stability. WTAP, Wilms’ tumor 1-associated protein; eIF3, eukaryotic initiation factor 3; *YTHDF*, YT521-B homology domain family; *YTHDC*, YT521-B homology domain-containing protein; IGF2BP, insulin-like growth factor 2 mRNA-binding protein; HNRNP, heterogeneous nuclear ribonucleoproteins; FTO-CTD, FTO C-terminal domain.

**Figure 2 cells-11-01516-f002:**
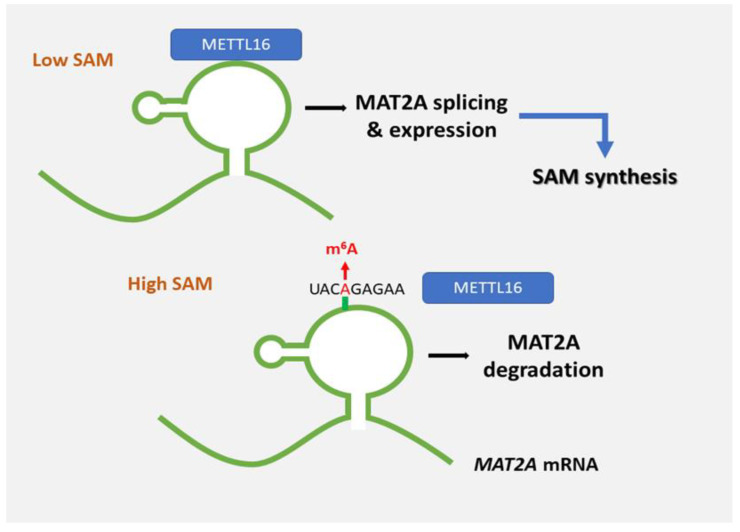
*METTL16* methylates a stem-loop structure in the 3′ untranslated region (UTR) of S-adenosyl methionine (SAM) synthase, methionine adenosyltransferase 2A (*MAT2A*). In SAM-repleted conditions, *MAT2A* is methylated and degraded. Conversely, in SAM-depleted conditions, *METTL16* induces splicing and expression of *MAT2A*.

**Figure 3 cells-11-01516-f003:**
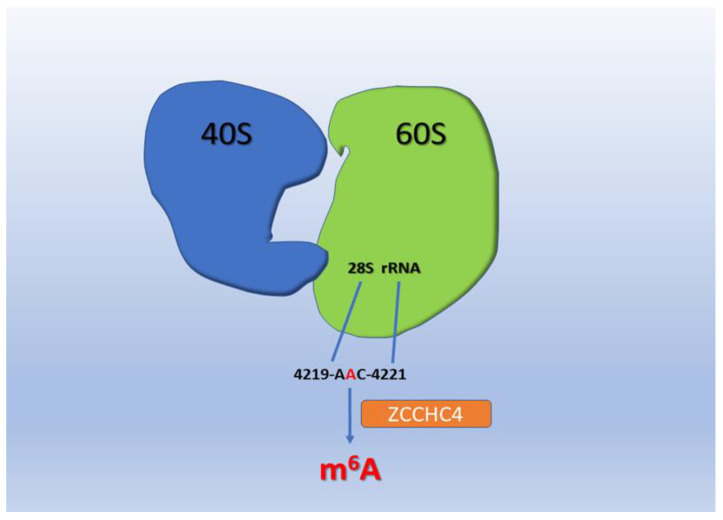
Methylation of A4220 in 28S ribosomal RNA (rRNA) by zinc finger CCHC domain-containing protein 4 (*ZCCHC4*) promotes ribosome assembly and translation.

**Figure 4 cells-11-01516-f004:**
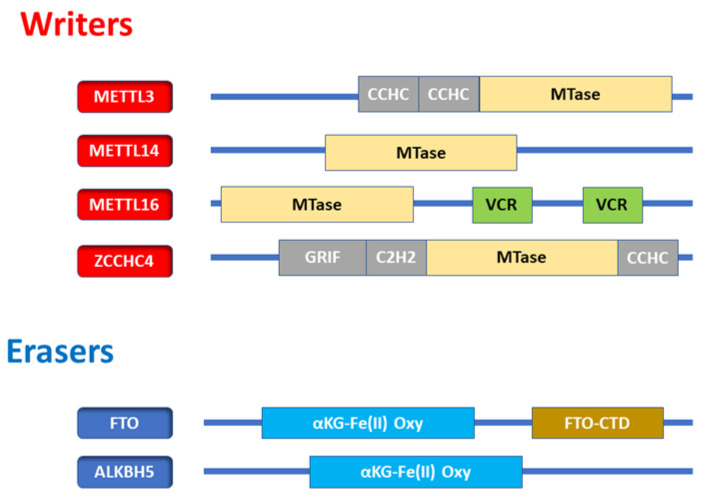
The domain composition of m^6^A enzymes. (Top, writers) m^6^A writers contain methyltransferase (MTase) domains. *METTL3* contains Cys-Cys-Cys-His (CCCH) zinc finger motifs. *METTL16* has two vertebrate-conserved regions (VCR).

## Data Availability

The data that support the findings of this study are available from the corresponding author upon reasonable request.

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
