# Peer review of "The Role of N6-Methyladenosine in the Promotion of Hepatoblastoma: A Critical Review"

_cells, 2022, doi:10.3390/cells11091516_

Round 1

Reviewer 1 Report

Auld et al. gave a critical review of the role of N6-methyladenosine in the promotion of hepatoblastoma, which may add to our knowledge in this field. However, the manuscript suffers several concerns that blowing away my enthusiasm to recommend it to be published at current version.

1. The abstract should be rewritten.

2. The authors spent too many paragraphs and figures in describing the form and functions of m6A, which should not be the key point for this paper.

3. The paper was not well organized.

Reviewer 2 Report

In this review Auld and colleagues discuss the role of m6A modification in hepatoblastoma. They briefly describe the m6A machinery and successively report the current knowledge of its involvement in hepatoblastoma progression. The studies reported are relevant and properly described. Despite this the evidence of a clear role of the m6a machinery in this cancer type is still limited therefore the review is somewhat speculative.

The introduction lacks discussion of specific aspects of the m6a modification machinery. And parts of the text are unclear and difficult to read. Many sentences make little sense and often the authors repeat themselves in different parts of the manuscript.

The manuscript cannot be accepted for publication in its current form, the authors should address the following major and minor points.

Major points

-FTO role as an m6A eraser is controversial since several studies showed that FTO is solely responsible for demethylation of m6Am on snoRNA 5’ cap. The authors should discuss this controversy

-The authors mention ZCCHC4as a ribosomal m6A writer but they do not mention METTL5 also involved in modifying ribosomal RNA

-The authors should discuss the topology of transcriptomic distribution of m6A. On which RNA types was it identified and at which specific positions within mRNAs?

-recently specific inhibitors of both m6A eraser and writers were developed, the authors should mention these potential new therapeutic agents in their introduction.

-in general the authors report the studies without critically discussing them.

-there are two Figure 2 legends within the manuscript.

Other points

-line 68   missing period

-line75    missing “14”

-line 86 m6A is the product of the reaction not a substrate

-lines 208-209 and 215-219 are redundant and should be removed

-lines 236-237 YTH proteins do not modify m6A but specifically bind the modification and mediate the downstream effects.

-line 264 and 279 “methylation of m6A” is misleading since m6a is the product of the methylation, not the substrate

-line 283 should include a reference to the reported paper

-line353 YTHDF2 is not a modifier of m6A but a reader

-line 417 the sentence makes little sense.

Reviewer 3 Report

The authors present a helpful review on the role of methylaminopurine in mammalian cancer cells. This is highly focussed, but certainly helpful for those involved in research on hepatoblastoma. 

The review is well-written, understandable and covers all important features of this type of base modification, associated with hepatoblastoma promotion in children. 

In this respect, the review is helpful, well-structured and certainly covering the most important literature in the field.

The reviewer has essentialy only two serious points of criticism. i. Whereas the review is helpful for experts in this area of research, it is not explicitly helpful for those, interested in starting research on pediatric hepatoblastoma or those - and this could in the end be the major audience - interested in the role of purine methylation on the whole.  

Thus, the reviewer insistently recommends to modify the manuscript towards this aim. Please provide a broadend introductory part, including comprehensively the effects of changes in adenine methylation patterns generally in eukaryotes. 

ii. The reviewer also recommends to place the important chapter on detecting and measuring RNA modifications earlier in the review. Discussing the technology earlier could motivate scientists to think seriously about similar studies in different systems.

On the whole, the authors present a reasonable addition to understanding RNA modifications that should be published after including improvement along the recommendations  above. 

Round 2

Reviewer 1 Report

1. The abstract did not seem meaningful with just a few words changed. 

2. I insist that the authors should delete some unimportant paragraphs and figures for the form and function of m6A. One figure is enough for this purpose.

3. The authors may add some figures to illustrate the functions of m6A in promotion of hepatoblastoma.

4. As the authors stated in the abstract, m6A-associated genes appear prominently in HB. What are those genes? What does m6A-associated genes mean? m6A-related genes? m6A-modified genes? Please make these clear.

5. The structure and flow are not as good as to be accepted. For example, when you describe key genetic players in HB, you should introduce mutated genes, m6A-driving genes (e.g. m6A readers, writers, and erasers), m6A-modified genes, and upstream or downstream molecules and pathways, in an order, instead of just listing them as you like.

6. Why “4. The Impact of m6A on Tissue Development” was listed as one of the parallel parts?  

7. In Diagnostic and Prognostic Biomarkers part, what about m6A-modified genes? Why did the authors spent a whole paragraph to describe tumor suppressive gene AX1N1, methylated genes, and CHKA? Were they m6A-modified or -related? For a review article, the timeliness, the breadth and accuracy of the discussion are very important.

Reviewer 2 Report

My major and minor criticism were successfully addressed in ther revised manuscript.